# Epidemiological Characteristics of a COVID-19 Outbreak in a Psychiatric Hospital in Chung-buk

**DOI:** 10.3390/healthcare11162332

**Published:** 2023-08-18

**Authors:** Se-Hyuk Jang, Young-Joon Park, Ji-Joo Lee, Woo-Jin Jung

**Affiliations:** Korea Disease Control and Prevention Agency (KDCA), Heungdeok-gu, Cheongju-si 28159, Republic of Korea; elliem01@korea.kr (S.-H.J.); pahmun@korea.kr (Y.-J.P.); jijoolee14@korea.kr (J.-J.L.)

**Keywords:** COVID-19, outbreak, psychiatric hospital

## Abstract

This study investigated the causes and risks for infection spread in three psychiatric hospitals in Chung-buk, South Korea, to strategize measures to block transmission and prevent a large-scale epidemic. From December 2020 to January 2021, 358 inpatients of Psychiatric Hospitals A, B, and C were enrolled to identify the epidemiological characteristics of confirmed patients. Epidemic curves and propagation relationships were constructed and a genotype analysis was conducted. The index case inpatient from Hospital A transmitted the infection to patients in Hospitals B and C; the infection was confirmed in 47, 193, and 118 patients in Hospitals A, B, and C, respectively. The patient characteristics hampered communication and the close identification of symptom onset. The incidence rate was 10 (2.9%) among employees and 348 (35.8%) among inpatients. The relative risk was 12.1 (95% CI: 6.6–22.5) times higher among inpatients than employees. Next-generation sequencing confirmed the probable infection source as a genotype identical to that of two different outbreaks, although the infection spread was undetermined. Direct risk factors emerged from patient characteristics, wherein cohort isolation was meaningless due to uncontrolled communication. Indirect risk factors included hospital-specific problems due to external factors (non-patient system deficiencies or employee negligence). Prior inspections, a confirmation of non-infection, and institutional emergent measures are needed.

## 1. Introduction

Coronavirus disease 2019 (COVID-19), caused by the severe acute respiratory syndrome coronavirus 2 (SARS-CoV-2), a member of the Coronaviridae family, is a highly contagious respiratory infection transmitted via respiratory routes, such as droplets, and contact with the virus [1]. First identified in the People’s Republic of China in December 2019 [2,3], COVID-19 has since achieved a pandemic status by infiltrating all areas of the globe [4]. By 12 April 2023, the disease accounted for 762,791,152 confirmed cases and 6,897,025 deaths worldwide [4]. South Korea reported its first case on 20 January 2020, and by 12 April 2023, the disease had occurred in approximately 60% of the total population (31,009,261 individuals), leading to 34,386 deaths (fatality rate: 0.11%) [5].

Following the first confirmed COVID-19 case in South Korea, the alert level for the infectious disease crisis was escalated to “caution”, then “warning” on 27 January 2020, and finally to “severe”, the highest level, on 23 February 2020 [6,7]. Despite vaccination efforts that commenced in 2021 [8], preventive strategies remain essential until vaccines or treatments become universally accessible. Thus, a social distancing policy was enforced in tandem with an emphasis on individual-level preventive measures, such as mask wearing and hand hygiene.

Psychiatric hospital patients often resist isolation due to a lack of insight into their condition [9]. Consequently, securing their cooperation can be particularly challenging. Prior to the COVID-19 outbreak at the Chung-buk psychiatric hospital, similar outbreaks have occurred worldwide. Several cases were reported outside of South Korea in the early days of COVID-19, including in China [10,11], the United States [12], and Italy [13,14], as well as in South Korea [10]. This possibility was starkly highlighted in the February 2020 COVID-19 outbreak at Daenam Hospital in Cheongdo, Gyeongsangbuk-do. Within 16 days of diagnosing the first patient on 7 February, 114 individuals were infected, of which 88.6% were inpatients who had been housed in shared rooms [15,16].

A policy enacted on 17 December 2020 mandated PCR testing prior to admitting patients to medical institutions [17]. However, given the 6 h minimum that could extend to a 1- to 2-day typical turnaround time for results with the COVID-19 screening kits approved for emergency use in South Korea [18], the management of suspected cases in emergencies or while awaiting test results proved challenging. In response to these challenges, following COVID-19 outbreaks in three psychiatric hospitals in Chung-buk, an epidemiological investigation was conducted in December 2020 [19,20]. The objective was to examine the risks associated with quarantine policy non-compliance, investigate institutional issues at psychiatric hospitals, and devise preventive strategies against infectious diseases. The investigation uncovered the first case of a multi-institutional, psychiatric-hospital-related epidemic in South Korea, which led to a significant outbreak in three different psychiatric hospitals [10,11].

The primary objective of this study was to conduct an epidemiological investigation and risk assessment to identify the characteristics of the psychiatric hospitals wherein the epidemic emerged [12,21,22]. This study was aimed at identifying the key causes and risk factors of transmission that could help devise strategies to manage future similar large-scale outbreaks in psychiatric hospitals by mitigating the risk of infectious disease transmission during patient transfer.

## 2. Materials and Methods

### 2.1. Outbreak Recognition

On 15 December 2020, five inpatients were transferred from Psychiatric Hospital A to Psychiatric Hospital B, wherein they underwent COVID-19 screening. Two of these patients tested positive for the virus. In the 2 days preceding this confirmation, six additional patients, who had shared a room with the two positively diagnosed individuals, also tested positive, indicating an outbreak in Hospital B. Consequently, an on-site epidemiological investigation was initiated on 17 December.

### 2.2. Hospital Information

Hospital A is a three-story structure with a total capacity of 170 beds. The first floor is designated for outpatients and the second and third floors house inpatient rooms; the third floor is reserved for closed wards. The hospital’s clinical departments encompass a wide range of specialties, including internal medicine, surgery, neurosurgery, orthopedic surgery, anesthesiology, neuropsychiatry, radiology, obstetrics and gynecology, dentistry, and an emergency room. Hospital B is significantly larger, comprising three separate buildings with 14 wards, and possesses the capacity to accommodate 615 patients and 173 employees. This hospital primarily functions as a long-term care facility for neuropsychiatric patients. Similarly, Hospital C consists of three buildings, providing a total of 256 beds. Like Hospital B, it is operated as a long-term care facility catering to the needs of neuropsychiatric patients.

### 2.3. Epidemiological Investigations

The initial step involved an epidemiological survey conducted by the public health center of the city where each psychiatric hospital was located, in collaboration with the Korea Centers for Disease Control and Prevention. Hospital A differed from Hospitals B and C in its accommodation setup, with six beds per room as opposed to up to 20 patients housed in a room in bunk-bed style in the latter two facilities. Hospital B’s and Hospital C’s patient management faced significant challenges. Patients often did not heed the staff’s instructions regarding infection control measures, such as mask and glove use. Furthermore, owing to the specific needs of psychiatric hospitals, the windows were made of acrylic instead of glass to prevent patient escapes, and were not designed to open, thereby impeding regular ventilation.

Secondly, sample collection for testing was performed by the medical staff of each psychiatric hospital. This approach was taken to minimize the patient distress that could be caused by unfamiliar outsiders. Samples were subsequently analyzed via polymerase chain reaction (PCR) at the Korea Institute of Health and Environmental Research.

Thirdly, basic patient information was provided by each hospital to the public health center. This information was then reported via the COVID-19 information management system operated by the Korea Centers for Disease Control and Prevention. However, to ensure the protection of personal information, only data pertaining to confirmed cases were collected, whereas information related to unconfirmed patients was excluded.

Lastly, next-generation sequencing (NGS) testing was conducted in partnership with the Novel Pathogen Analysis Division of the National Institute of Health within the KCDC. This strategy was employed to compare the genotype of the index patient or the patient with the earliest onset of symptoms among confirmed cases, with the genotype of index patients in other regions or case clusters. This process facilitated the identification of infection routes by matching the genotype of the index patient with the region or individual from whom the outbreak originated.

### 2.4. Case Definition

A confirmed case of SARS-CoV-2 was defined as a case in which a SARS-CoV-2 infection was confirmed by PCR testing and reported to the COVID-19 information management system. Confirmed cases of COVID-19 in psychiatric hospitals in Chung-buk were defined as workers, inpatients, and family members associated with Psychiatric Hospitals A, B, or C in Chung-buk who tested positive for SARS-CoV-2 by RT-PCR during the epidemic period (symptom-onset date to epidemic-ending date) and were reported to the COVID-19 information management system.

### 2.5. Data Collection and Analysis

The index case was a psychiatric inpatient at Hospital A who was transferred from Hospital B on 7 December. Considering the 10-day incubation period of COVID-19 in asymptomatic patients [23,24], the risk period of the first exposure was determined to be from 5 December to 16, 2020. An epidemiological investigation revealed symptom onset in the index case on 9 December and a symptomatic inpatient on 5 December; thus, the exposure risk period was re-estimated to be from the end of November to the beginning of December. As the exact exposure date was difficult to identify, the date of symptom onset of the index case (9 December) and the date of transfer (7 December) were used to identify the employees, inpatients, and family members of employees of the three psychiatric hospitals as subjects of investigation. As additional confirmed cases occurred in other unrelated wards within each hospital, all employees and inpatients of the hospital buildings were included as subjects of investigation. Based on the list submitted by the three hospitals, there were a total of 1310 subjects: 248 at Hospital A, 766 at Hospital B, and 296 at Hospital C. Table 1 presents a detailed number of subjects according to the hospital.

Epidemic epidemiological curves were analyzed based on the general characteristics of confirmed cases, including sex, age, and place, as well as the date of symptom onset (replaced by the date of confirmation for asymptomatic patients), to describe the epidemic pattern and estimated exposure period. In addition, for analyzing the in-hospital transmission, the participants were divided into confirmed and non-confirmed cases, and the incidence rate and risk factors of each facility for the SARS-CoV-2 infection were analyzed using a retrospective cohort study design.

For the statistical analysis, descriptive statistics were used for general characteristics by sex, age, and facility, and the relative risk (RR) and 95% confidence intervals (CIs) were calculated to evaluate the risk. A frequency analysis and descriptive statistical analyses were conducted in Microsoft Excel 2019 (Microsoft Corporation, Redmond, WA, USA), and EpiInfo v5.5.3 (US Centers for Disease Control and Prevention, Atlanta, GA, USA) was used to calculate the RR and statistical significance. *p* < 0.05 was considered to indicate statistical significance.

When an epidemiological association between patients was estimated through epidemiological investigations, residual samples from the diagnostic testing institution at the time of confirmation were obtained and next-generation sequencing (NGS) was performed. The results of the NGS analysis were compared by single-nucleotide polymorphisms (SNPs) to confirm the genetic relatedness between the infectious agent and additional transmission and to reinforce the evidence base for the epidemiological association.

## 3. Results

### 3.1. General Characteristics

From the three psychiatric facilities in Chung-buk, 358 confirmed SARS-CoV-2 cases were identified, with 325 (90.8%) male and 33 (9.2%) female infectees. The age distribution of the cases was as follows: 2 (0.6%) in their 10s, 20 (5.5%) in their 20s, 43 (12.1%) in their 30s, 45 (12.6%) in their 40s, 97 (27.0%) in their 50s, and 151 (42.2%) aged 60 and over (Table 1).

Hospital A reported 47 confirmed cases, including 34 (72.3%) male and 13 (27.7%) female patients. The majority of these patients were aged 60 years or more (n = 23; 48.9%). Hospital B had 193 confirmed cases, of which 173 (89.6%) were male and 20 (10.4%) were female, again with the majority in the 60s-or-older age group (n = 85; 44.1%). In Hospital C in Eumseong, all 118 confirmed cases were men, with 40 (33.9%) in their 50s and 43 (36.4%) in their 60s or older. In contrast to Hospitals A and B, Hospital C saw a substantial number of confirmed cases in two age groups, the 50s group and the 60s-or-older group. Overall, the majority of confirmed cases in all three psychiatric hospitals involved male patients, and most confirmed cases were identified as individuals aged 60 years or more (Table 1).

On evaluating the relationship among the confirmed cases, we found that, out of the 358 total cases, 10 (2.8%) were hospital employees: two from Hospital A and eight from Hospital B, but none from Hospital C. When stratified by age, the majority of infections among employees occurred in individuals aged 60 years or more (n = 6; 60.0%). The patient population consisted of 317 (91.1%) male and 31 (8.9%) female individuals, with a high incidence rate among males. When analyzed by age, the incidence rate was high among those in their 50s and 60s or older in all three psychiatric hospitals. The total number of patients in the three hospitals was 971, with 760 males (78.3%) and 211 females (21.7%). There were 317 confirmed cases among males, representing 41.7% of all patients, and 31 confirmed cases among females, representing 14.7%.

Overall, by relationship, gender, and age group, SARS-CoV-2 infections were more prevalent among inpatients, males, and those in their 50s and 60s or older. Although not statistically verified, one participant with a confirmed SARS-CoV-2 infection mentioned that outdoor activities, such as exercise classes and breaks (including smoking breaks), were prohibited due to the spread of SARS-CoV-2. Consequently, large numbers of people would gather in small spaces to smoke, and this potentially contributed to the transmission of the infection.

### 3.2. Incidence Rate and RR (Relative Risk)

The incidence rate of confirmed patients in the three psychiatric hospitals was 27.3% (358 out of 1310 inpatient cases) and, facility-wise, was 19.0% (47 out of 248 inpatient cases) in Hospital A, 25.2% (193 out of 766 inpatient cases) in Hospital B, and 39.9% (118 out of 296 inpatient cases) in Hospital C. The incidence rate was 2.9% (10 out of 339 inpatient cases) for employees and 35.8% (348 out of 971 inpatient cases) for inpatients (Table 2). 

The RR revealed that, overall, the SARS-CoV-2 incidence rate for inpatients was 12.1 times higher than that for employees (*p* < 0.001), which was significant. In Hospitals A, B, and C, the SARS-CoV-2 incidence rate for inpatients was 22.1, 5.7, and 65.6 times higher, respectively, than that for employees (all *p* < 0.05, Table 2). 

### 3.3. Epidemic Curve

The epidemic curve related to cluster outbreaks in the three psychiatric hospitals in Chung-buk commenced on 5 December 2020, when the earliest symptoms were confirmed; ended on 6 February, exactly 14 days after the last confirmed case was identified on 23 January; and presented a typical epidemic curve for a respiratory infection. Based on the prevalence by relationship during the incubation period, the infection started among inpatients and progressed to employees and other inpatients. In addition, based on the date of symptom onset, we can assume that there was a nosocomial exposure from about 5 December and that the patient may have spread the infection to other patients from that time. However, from the time of nosocomial transmission, as a measure for hospitals, PCR tests were conducted every 3 days as a preemptive test, and many cases were confirmed positive; therefore, it was difficult to clearly identify when the symptoms occurred and, in such cases, asymptomatic patients were considered (Figure 1). 

### 3.4. Epidemiological Findings

The psychiatric ward of Hospital A, where the patient with SARS-CoV-2 was hospitalized, is a cooperative hospital that treats physical illnesses of psychiatric patients and has a high density of inpatients. Considering the high-risk facility environment, we evaluated contacts with the confirmed SARS-CoV-2 patient. Based on the estimated exposure period from 5 to 26 December 2020, all residents on the second and third floors of the psychiatric ward, where the confirmed SARS-CoV-2 patient was located, were tested for a SARS-CoV-2 infection. Wards 2 and 3 were operated without clear boundaries between floors based on the clinical status or severity of patients, which may have resulted in additional exposures. The testing confirmed 20 additional cases (2 staff and 20 inpatients (2 deaths)) on the third-floor closed psychiatric ward, bringing the total number of confirmed cases to 22 as of 18 December 2020. 

Despite the PCR-based confirmation of a SARS-CoV-2 infection in the index patient at Hospital A, all five patients awaiting transfer, including the index patient, were transferred to Hospital B as scheduled because they complained of fatigue and it was expected to take much longer for PCR test results to be reported. Approximately 6 h after the transfer, a PCR-based confirmation of SARS-CoV-2 infection was obtained and the patients were transferred from Hospital B back to Hospital A. Given the exposure of patients at Hospital B, SARS-CoV-2 testing was performed on all individuals in Wards 9 and 10 of Hospital B, where the two index cases had stayed, resulting in six additional cases (18 December).

Due to the outbreak in both hospitals due to the transfer and re-transfer of patients from Hospital A and Hospital B, targeted testing was conducted in 13 psychiatric hospitals and long-term care facilities in Chung-buk, Sejong-si, and Gyeonggi-do (seven, one, and five hospitals, respectively), including Hospital A, which had frequent patient and inpatient exchanges, and other hospitals to which patients from Hospital A were transferred. A total of 38 patients from Hospital A were transferred to the 13 institutions, and one additional case was confirmed (24 December) at Hospital C, where the index patient was hospitalized.

The outbreak was officially named the “Chung-buk psychiatric hospital-associated outbreak”, as it occurred in three local governments, three hospitals, and a psychiatric hospital in the province. Hospitals B and C were classified as “subgroups”, and a plan for management and follow-up was developed and implemented, taking into account the high-risk facility environment of the hospitals and the possibility of a future large-scale outbreak. The plan includes a “patient management plan” for transfer to the National Mental Health Center (M Center), a “hospital exposure management plan”, a “shared isolation facility (bed) management plan”, a “hospital personnel management plan”, and a “hospital infection control plan”. The three hospitals where the outbreak began were ordered to close, and monitoring tests were conducted at 3-day intervals from the last exposure. On the 13th, the day after the last exposure, testing was conducted for staff in wards without cases in the same hospital. We also asked seven hospitals in Chung-buk, one in Sejong-si, and five in Gyeonggi-do without cases to conduct periodic monitoring tests. We followed up with additional cases in the three hospitals and confirmed 91 cases in Hospital B (23 December) and 21 cases in Hospital C (24 December) through monitoring tests. We followed up with Hospital B, which had a large number of confirmed cases, as it was necessary to transfer patients to the National Psychiatric Center due to the nature of psychiatric patients who were not general COVID-19 cases. However, the number of beds at the National Psychiatric Center was absolutely insufficient, with only 22 beds as of 18 December. In addition, due to the nature of psychiatric hospitals, ventilation was difficult, and it was impossible to enforce social distancing between inpatients due to high-density living. The risk was so high that it was pointless to even assess the risk, as we could only expect compliance with quarantine measures among staff.

The specific reasons were as follows: Psychiatric hospitals and wards had barred acrylic windows to prevent inpatients from escaping, and the windows were secured so that they could not be opened. In terms of density, some patients were able to communicate, but most were not, and they were all packed together. It was not possible to separate areas for those with a confirmed SARS-CoV-2 infection from those without, as the inpatients could become unruly while under control. It seemed impossible to maintain social distancing between patients or reduce the density. Therefore, we decided to address these issues through a number of measures.

### 3.5. Infection Diagram

On 15 December, the first infected patient(index case) transferred to Hospital B, infected 193 patients in Hospital B. A patient had contact with the first infected patient in Hospital A transferred to Hospital C on 24 December, infecting 118 patients in Hospital C. A staff of Hospital B then transmits to the staff’s family on January 10 (Figure 2).

### 3.6. NGS Test Results

The clade was of the GH type, with a lineage from the GSCC family, and this viral strain 100% matched the genotype in the “Cluster Related to Religious Facilities in Church, Chungcheongnam-do (Chung-nam)” and the “Cluster Related to the Detention Center in Prison, Seoul”. Although the association of the two clusters, such as the contact history, could not be confirmed, as far as it could be estimated, the “Cluster Related to Religious Facilities in Church, Chung-nam” occurred approximately 1 week before the index case in this cluster (Figure 3).

### 3.7. Measures

This multifaceted approach to the management of the SARS-CoV-2 outbreak in the psychiatric hospitals involved several strategies:**Patient management plan**: Patients were categorized into three groups: confirmed (infected), exposed, and unexposed. The confirmed and exposed individuals were grouped together, while the unexposed individuals were separated to limit the further spread of the virus [25].**Hospital exposure management plan**: Every three days, all individuals in the hospital underwent testing for SARS-CoV-2. People with symptoms and those already confirmed to have the virus were tested even more frequently.**Public quarantine facility management plan**: For individuals who were exposed, but not confirmed to have the virus, it was deemed unsafe for them to stay in the psychiatric hospital due to the high-risk environment. These individuals were transferred to national psychiatric hospitals. If these hospitals reached capacity, some spaces in the hospitals were designated as pre-preparation rooms.**Hospital workforce management plan**: Staff who had been exposed to confirmed cases stayed in the confirmed case area for two weeks to provide care and implement infection control measures. If they tested negative after two weeks, they were rotated out and replaced by other staff members to ensure a streamlined workflow and preparation for a prolonged outbreak [26,27].**Hospital infection control plan**: Hospital staff were trained on cohort isolation, the correct usage of personal protective equipment (PPE), access control, transportation, and waste management. These measures were managed by the health center.**National psychiatric hospital plan**: Due to the continued increase in SARS-CoV-2 infections among psychiatric patients and the limited bed capacity, there was a need for a dedicated facility. Based on an agreement among the Ministry of Health and Welfare, the Chungcheongbuk-do office, Hospital B was converted into a specialized COVID-19 hospital for mildly mentally ill patients. This allowed for the treatment of COVID-19 patients from other psychiatric hospitals in the country at Hospital B. The policy was implemented to offset some of the financial losses incurred by Hospital B during the outbreak, thereby benefiting both the hospital and government agencies.

## 4. Discussion

This is the first report of a large-scale psychiatric hospital outbreak that occurred in a multi-institutional setting with all positive patients in a psychiatric hospital in Chung-buk. In Korean medical institutions, PCR tests were performed before admission (or transfer). However, owing to the long time before the results were confirmed, the hospitalized patients were not separated, and in-hospital transmission occurred. It was confirmed that the infection spread to Hospital A, where the index patient was initially hospitalized; to Hospital B, where he was transferred; and to Hospital C through an epidemiological investigation. Psychiatric hospitals are a group with special characteristics that differ from general medical institutions, and several risk factors have been identified that can lead to large-scale outbreaks.

First, the use of “multi-person rooms”, a structure wherein many people are gathered together in a large space and live on the floor, inevitably leads to more contact than beds [25]. Second, due to the “sealed environment” and patients trying to escape from the psychiatric hospital, the windows were made of acrylic rather than glass, and the windows could not be opened, so ventilation was not possible [25]. Since this is a problem that cannot be changed due to the nature of a psychiatric hospital, we installed a negative pressure device on the ceiling that was out of the reach of patients to enable ventilation. Third, we appointed representatives of the “uncontrolled patients”, those who can communicate and cooperate with the staff, to manage the patients who are difficult to communicate with through them [28]. This is a very important issue, as a patient who is unable to communicate may have a fever and might not be able to express that they are sick, which could cause respiratory droplet-based infection spread to other patients. Fourth, “exposure through consumption” was an issue. Although contact with the outside world was prevented, there was a risk of exposure through frequent outings between patients and their own consumption, which could not be enforced. Therefore, contact with people from other wards was prohibited, and the outing time was divided into each ward to minimize the time to meet with patients from other wards [29].

The epidemiological investigation had several limitations. First, it was not possible to interview the patients and obtain detailed information, which made it difficult to find the route of infection. As an alternative, we tried to link the cases to other clusters through NGS testing, but this was not possible because there was no way to identify the contact between the two groups in the event of another large-scale outbreak. Efforts should be made to obtain and analyze the genotypes of individuals that can be collected to identify the source of the infection. Second, as the cohort quarantine was implemented, negative-pressure rooms were required to be installed, but due to the lack of a budget for the hospital, the installation of negative-pressure rooms was delayed. However, it was possible to compensate through budget support in consultation with the Chung-buk province and the Ministry of Health and Welfare. Third, only univariate statistics were used in the analysis, and a multivariate analysis reflecting in-depth and diverse variables could not be conducted. This part was a limitation of the information collection; information could be collected through interviews with patients, but interviews were not possible, and we tried to check and reflect on the medical records as an alternative method, but the medical records did not include epidemiological variables, so they were difficult to check, which remained a limitation of the study.

Despite these limitations, this study was able to identify realistic major risk factors that may occur in the field by visiting the sites of three medical centers, and all cases were examined extensively to determine the incidence and mortality of all cases involved. In addition, all negative patients were tested every three days to check for further outbreaks and minimize nosocomial transmission. 

Through this case, it was confirmed that infectious diseases that are introduced into a hospital, due to the structure of closed wards and uncontrolled psychiatric patients, can spread quickly and on a large scale. Therefore, strategies for infection control in psychiatric hospitals should focus on blocking the introduction of infectious diseases at the hospital level [30]. For this purpose, first, infection control education for hospital workers should be provided. Second, periodic testing should be performed. Third, patients should be separated from pre-admission testing until the results are confirmed. Fourth, restrictions on visitors can prevent infectious diseases from entering the hospital [31].

In addition, the central government, with the cooperation of other organizations, operated dedicated hospitals for the treatment of psychiatric COVID-19 patients nationwide, reducing the burden on frontline medical institutions and enabling the systematic management of confirmed cases. These measures are expected to be utilized to obtain basic data to prepare healthcare policy measures that can be implemented temporarily in disaster situations.

The implications of this study include the difficulty of infection control for individuals due to the nature of psychiatric hospitals and the limitations of cohort isolation and management due to communication problems, but nevertheless, we found a way to cohort-isolate the hospital itself and separate positive and negative patients by floor when a detailed isolation was not possible. Second, when a large number of people were exposed, we were able to detect cases in advance by conducting a full screening every three days; third, we provided reinforcement training on infection control (how to isolate cohorts, how to wear protective equipment, and how to separate exposed and unexposed people) to workers who lacked awareness of the infection so that the hospital could manage itself; and fourth, we found a strategy to turn the hospital into a nationally managed hospital by providing a budget, manpower, and materials to the hospital in case of an emergency.

## 5. Conclusions

This study describes the process of an epidemiological investigation during an outbreak of respiratory infectious diseases in psychiatric hospitals and proposes strategies for preventing the spread of transmission and managing confirmed cases based on the findings. According to the results of the epidemiological survey, during the outbreak of COVID-19 in psychiatric hospitals, the use of multi-person rooms, narrow spaces without ventilation, and frequent close contact necessitated by the patient characteristics led to rapid transmission, and management was difficult due to a lack of compliance with precautions, not wearing protective equipment such as masks, and uncontrollable and unrecognized pain. The management plan for psychiatric hospitals should be to quickly conduct screening tests to distinguish between confirmed and non-confirmed cases; at the same time, set up exposed and non-exposed areas to minimize exposure to others by separating places; consider strategies to access facilities, rather than individuals; and improve policies that can be applied exceptionally in disaster situations.

## Figures and Tables

**Figure 1 healthcare-11-02332-f001:**
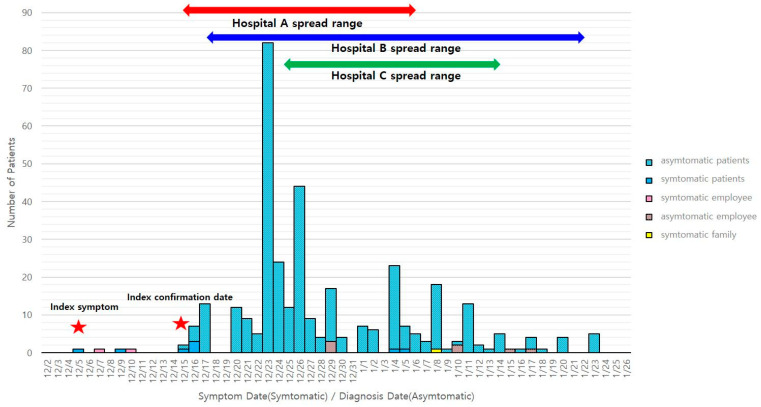
Epidemic curve for COVID-19 infections in patients, employees, and families of the three psychiatric hospitals in Chung-buk.

**Figure 2 healthcare-11-02332-f002:**
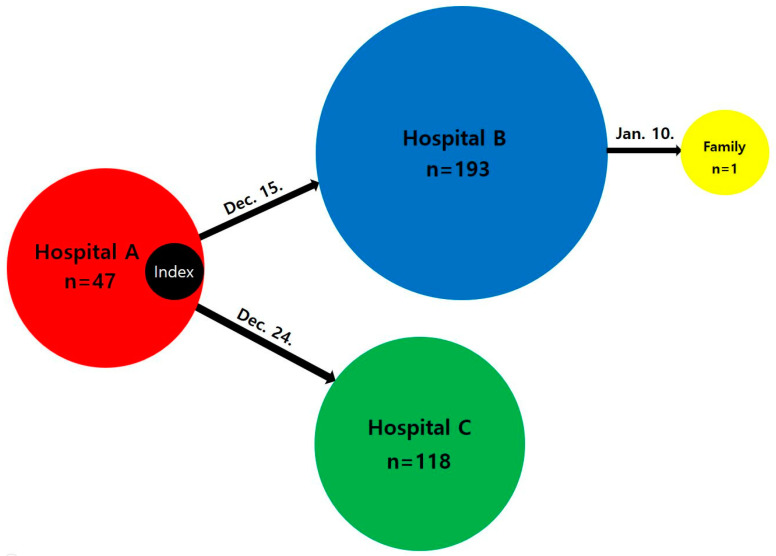
The Infection diagram from Hospital A to Hospital B, C and Family.

**Figure 3 healthcare-11-02332-f003:**
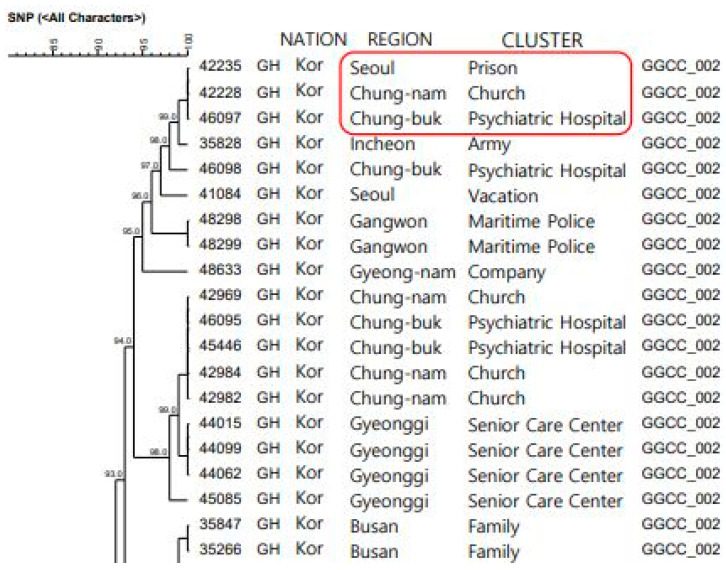
NGS test results related to the psychiatric hospitals in Chung-buk.

**Table 1 healthcare-11-02332-t001:** Demographics of confirmed SARS-CoV-2 infection cases at three psychiatric hospitals in Chung-buk.

	Total		Hospital A		Hospital B		Hospital C	
	n (%)		n (%)		n (%)		n (%)	
Total	358	(100.0)	47	(100.0)	193	(100.0)	118	(100.0)
Sex								
Male	325	(90.8)	34	(72.3)	173	(89.6)	118	(100.0)
Female	33	(9.2)	13	(27.7)	20	(10.4)	0	(0.0)
Age, decade								
10s	2	(0.6)	1	(2.1)	1	(0.5)	0	(0.0)
20s	20	(5.5)	2	(4.3)	16	(8.3)	2	(1.7)
30s	43	(12.1)	5	(10.6)	24	(12.4)	14	(11.9)
40s	45	(12.6)	2	(4.3)	24	(12.4)	19	(16.1)
50s	97	(27.0)	14	(29.8)	43	(22.3)	40	(33.9)
≥60s	151	(42.2)	23	(48.9)	85	(44.1)	43	(36.4)
Status								
Employee	10	(2.8)	2	(4.3)	8	(4.1)	0	(0.0)
Inpatient	348	(97.2)	45	(95.7)	185	(95.9)	118	(100.0)

**Table 2 healthcare-11-02332-t002:** Incidence and relative risk (RR) of SARS-CoV-2 infection in three psychiatric hospitals.

	Classification	Positive	Negative	Incidence Rate	RR (95% CI) *p*-Value
Total	Employee	10	329	2.9%	Reference
	Inpatient	348	623	35.8%	12.1 (6.6–22.5) *p* < 0.05
Hospital A	Employee	2	121	1.6%	Reference
	Inpatient	45	80	36.0%	22.1 (5.5–89.3) *p* < 0.05
Hospital B **	Employee	8	144	5.3%	Reference
	Inpatient	185	429	30.1%	5.7 (2.9–11.4) *p* < 0.05
Hospital C	Employee	0 *	64	0.0%	Reference
	Inpatient	118	114	50.9%	65.6 (4.1–1040.8) *p* < 0.05

* Calculated by replacing 0 with 0.5. ** Calculated by including one family member related to Hospital B as an employee.

## Data Availability

Not applicable.

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
