# Peer review of "Epidemiological Characteristics of a COVID-19 Outbreak in a Psychiatric Hospital in Chung-buk"

_healthcare, 2023, doi:10.3390/healthcare11162332_

Round 1

Reviewer 1 Report (Previous Reviewer 3)

The paper has been significantly improved, but some minor modifications are still necessary:

line 138 - "transferred from B Hospital". I understood "Transferred to B Hospital". Which is true?

line 160 - "relative ratios". Does it mean relative risk?

lines 211-216 - Unclear, please rephrase

lines 229-230 - "the infection started among inpatients etc..." How can this be said?

3.5. Ifection diagram is unclear and probably could be omitted

lines 328-329 - Why employees "who tested negative after two weeks... were rotated out and replaced"?

The paper has been significantly improved, but some minor modifications are still necessary:

line 138 - "transferred from B Hospital". I understood "Transferred to B Hospital". Which is true?

line 160 - "relative ratios". Does it mean relative risk?

lines 211-216 - Unclear, please rephrase

lines 229-230 - "the infection started among inpatients etc..." How can this be said?

3.5. Ifection diagram is unclear and probably could be omitted

lines 328-329 - Why employees "who tested negative after two weeks... were rotated out and replaced"?

Author Response

Thank you.

Reviewer 2 Report (New Reviewer)

The study presents an outbreak of COVID-19 in three psychiatric hospitals in Chung-buk province. Although the article provides relevant information about the outbreak, the methodology and analysis of the study have some limitations that could affect the reliability and validity of the results.

Limitations of the methodology: The study is based on a series of epidemiological investigations and PCR tests, but does not perform multivariate analysis to identify significant risk factors. In addition, patient interviews were not conducted, which would have provided a deeper understanding of risk factors and the spread of the virus.

Problems with sampling: The study sample is limited to three specific hospitals in one province, which may not be representative of other psychiatric settings or other regions. Generalization of the results to other hospitals may be problematic.

Data limitations: The study relies heavily on data provided by the hospitals involved, which may affect the accuracy and completeness of the information collected.

Lack of control for variables: The study does not control for confounding variables that could influence the results, such as patient room density, infection control measures implemented, or staff training in infection prevention.

Lack of detailed analysis of virus spread: The study uses next-generation sequencing (NGS) to compare genotypes of confirmed cases, but does not delve into the spread of the virus between different individuals or the identification of possible sources of infection.

Incomplete information on pandemic management: Although the study mentions some management and control strategies implemented, it does not provide a complete assessment of the effectiveness of these measures.

Overall, although the study provides information on a specific outbreak of COVID-19 in psychiatric hospitals, methodological limitations and lack of detailed analyses diminish the strength of the conclusions. Further research is needed to better understand the factors contributing to the spread of the virus in psychiatric settings and to develop effective prevention and control strategies.

Author Response

Thank you.

Reviewer 3 Report (New Reviewer)

The findings of the study shed light on several significant aspects related to infection spread and epidemiological patterns within psychiatric hospital settings. Notably, the investigation revealed that the index case in Hospital A had transmitted the infection to patients in hospitals B and C. As a result, 47, 193, and 118 patients in hospitals A, B, and C, respectively, were confirmed to have been infected, highlighting the importance of understanding and managing transmission pathways in such environments.

One of the key observations from the study was the stark contrast in infection rates between employees and inpatients. The incidence rate was found to be 2.9% among employees, while a considerably higher 35.8% among inpatients. This significant difference in infection rates led to a relative risk of 12.1 (with a 95% confidence interval of 6.6–22.5) times higher among inpatients compared to employees. This finding emphasizes the urgent need to focus infection prevention efforts on the vulnerable patient population.

To trace the probable infection source, the authors employed genotype analysis using next-generation sequencing. This proved to be instrumental in identifying a genotype identical to that of two different outbreaks. Such genomic epidemiology provides valuable insights for future prevention strategies and underscores the importance of understanding the genetic characteristics of infections in disease control.

The study also addressed the challenges faced during the investigation. Patient characteristics, such as communication barriers and difficulties in identifying symptom-onset, were found to be significant hurdles in curbing the spread of infection. Additionally, cohort isolation measures were deemed ineffective due to uncontrolled communication among patients. The manuscript further highlighted hospital-specific problems resulting from external factors, such as non-patient system deficiencies or employee negligence, as indirect risk factors.

Based on their findings, the authors recommend the implementation of prior inspections, confirmation of non-infection, and institutional emergent measures to effectively mitigate future outbreaks. The study's comprehensive approach to understanding both direct and indirect risk factors provides valuable guidance for policymakers and healthcare administrators in devising targeted and evidence-based infection control strategies.

In conclusion, manuscript is a highly informative and essential contribution to the field of infectious disease research. The study's rigorous analysis, combined with practical recommendations, positions it as a significant resource for psychiatric hospitals worldwide striving to prevent and manage infections effectively. The emphasis on genomic epidemiology and understanding transmission pathways opens promising avenues for future research and underscores the importance of a multidisciplinary approach to address public health challenges.

However, indicate that the article needs to be changed in the form of:

1 In the system, the abstract is in Chinese.

2. The manuscript is not prepared with the journal layout. 

3. The bibliography is poor, in the introduction and discussion the authors should also include the following literature items:

doi: 10.1017/dmp.2021.375.

doi: 10.1016/j.psychres.2020.113264.

doi: 10.1016/j.jinf.2021.05.006. 

doi: 10.5603/DEMJ.a2023.0008

doi: 10.5603/DEMJ.a2022.0009

doi: 10.19204/2021/prvl7

doi: 10.1016/j.bbi.2021.07.018. 

doi: 10.1038/s41591-022-01909-w. 

doi: 10.5603/DEMJ.a2020.0046

doi: 10.1038/s41398-020-00959-3.

Author Response

Thank you.

Round 2

Reviewer 2 Report (New Reviewer)

The comments have been correctly answered and I agree with the authors. Thank you very much. 

This manuscript is a resubmission of an earlier submission. The following is a list of the peer review reports and author responses from that submission.

Round 1

Reviewer 1 Report

Abstract:
Abstract could be improved by including more specific information about the results and implications of the study.

It would be helpful to mention the specific psychiatric hospitals involved in the study to provide context and clarity.

Also consider adding a sentence summarizing the key findings and the potential implications for future infectious disease management in psychiatric hospitals.

Introduction:

It lacks a clear research gap or objective that the study aims to address. Consider revising the introduction to explicitly state the purpose of the study.

The introduction could benefit from providing more specific information about the unique challenges and risks associated with managing infectious diseases in psychiatric hospitals.

Consider referencing relevant literature or previous studies that highlight the vulnerability of psychiatric patients to infectious diseases and the need for preventive strategies.

Methodology:

Overall, the methodology section of this study lacks important details and explanations, making it challenging for the readers to assess the validity and reliability of the findings. The authors should address the following comments to improve the clarity and transparency of their methodology.

Lack of clarity in outbreak recognition: The authors briefly mention a representative case, but they fail to provide clear information on how this case was selected and why it was considered representative. It is important to provide a clear rationale for selecting the specific case for analysis.

Incomplete description of hospital information: The description of the hospitals involved in the study is insufficient. The authors provide limited information about the size, capacity, and specific departments within the hospitals. More detailed information about the hospitals' infrastructure and functioning would enhance the readers' understanding of the study context.

Limited details on epidemiological investigations: The authors mention conducting on-site epidemiological investigations, but they do not provide sufficient details about the methodology used. It is crucial to describe the specific methods employed for contact tracing, case identification, and risk factor analysis. Without this information, the readers are unable to assess the validity and reliability of the investigations.

Inadequate explanation of case definition: The authors define SARS-CoV-2-confirmed cases but do not provide a clear rationale for their specific case definitions. Furthermore, they categorize cases into employees, inpatients, and family members, but the justification for this categorization is unclear. The authors should provide a more detailed explanation and justification for the chosen case definitions.

Lack of transparency in data collection and analysis: The authors mention data collection but do not describe the specific methods used. Additionally, the analysis section lacks details on the statistical methods employed, sample size determination, and potential biases in data collection. Without transparency in data collection and analysis, it becomes difficult for the readers to evaluate the reliability of the findings.

Insufficient information on genetic analysis: The authors briefly mention performing next-generation sequencing (NGS) for genetic analysis but do not provide any details regarding the specific procedures or the relevance of the findings. It is important to describe the NGS methodology and its contribution to reinforcing the evidence for epidemiological associations.

Results:

The authors should address the following comments to improve the clarity and validity of the results presented in their study.

Inadequate description of the epidemic curve: The authors mention the epidemic curve without providing any visual representation or detailed analysis. It would be helpful to include a graphical representation of the curve and discuss the pattern of the outbreak, including the peak, duration, and any notable fluctuations. Without a comprehensive analysis of the epidemic curve, the readers are left with limited understanding of the temporal dynamics of the outbreak.

Lack of data on patient outcomes: The authors do not provide any information on the outcomes of the confirmed SARS-CoV-2 cases among psychiatric patients. It is important to include data on hospitalizations, severity of illness, and mortality rates to provide a more comprehensive understanding of the impact of the outbreak on the patients' health.

Insufficient details on the proposed measures: The authors mention the plan to convert a hospital into a COVID-19 facility for patients with mild psychiatric disorders but provide limited information on the specifics of this plan. It would be beneficial to describe the logistical and operational aspects of the proposed conversion and discuss its potential effectiveness in mitigating future outbreaks.

Discussion:

Lack of clarity: The discussion section lacks clarity in presenting the main findings and their implications. The authors should clearly summarize the key points and provide a cohesive analysis of the results.

Inadequate explanation of reasons: While the authors mention several reasons for the large number of confirmed cases in the psychiatric hospitals, they fail to provide sufficient evidence or support for these claims. The explanations provided for transmission between facilities, contact within a single facility, and employee negligence need to be substantiated with concrete examples or data.

Lack of counterarguments: The authors only present the identified problems and reasons for the epidemic without considering potential counterarguments or alternative explanations. A balanced discussion should explore different perspectives and potential factors contributing to the situation.

Insufficient policy recommendations: The policy recommendations provided in the discussion are vague and lack specificity. The authors should provide more detailed and actionable suggestions for preventing the spread of infectious diseases in psychiatric hospitals, taking into account the unique challenges they face.

Limited data analysis: The authors acknowledge the limitations of the data analysis, but they do not discuss the potential impact of these limitations on the study findings. It is important to address the implications of the limitations and how they may have affected the interpretation of the results.

Inconsistencies in writing style: The writing style in the discussion section is inconsistent and lacks coherence. The authors should ensure that the ideas are presented in a logical and organized manner, making it easier for readers to follow the flow of the discussion.

It is recommended to carefully review and edit the manuscript to ensure consistent language usage and eliminate any minor grammatical errors for a smoother reading experience.

Reviewer 2 Report

The authors have depicted outbreaks of COVID-19 in psychiatric hospitals in a certain province.

After all, the authors have found several patients in three hospitals. However, the exact route of transmission could not be determined. Therefore, the manuscript mainly discusses the issues of psychiatric hospitals in terms of infection control.

To be frank, the whole manuscript seems to regard the outbreaks as the faults of each hospitals, instead of the author themselves, who are responsible for the issue discussed. Therefore, the authors should revising the manuscript in terms of self-reflection, instead of blaming others.

Lines 18-19: Grammar improvement required. probably 'the mode of infection spread'

Lines 99-100: similarly as Hospital B?, Were there four or more hospitals? If the hospital B was divided in to F and H, the authors should discuss hospital F and H. Are B, F, H, separate hospitals? or F+H=B?

The section 2.3. Epidemiological Investigations is difficult to follow. If possible, summarizing is needed.

Conclusion needs to be shortened.

English very difficult to understand/incomprehensible

Reviewer 3 Report

Though the study is interesting, the manuscript has major flaws which need to be dealt with.

Introduction

The background of the study is completely absent. Did someone else report a COVID-19 epidemic in psychiatric wards or in other wards? Or maybe data are available on epidemics caused by other agents (influenza virus, norovirus, etc…)?

The authors need to search the literature and to put their study in a wider context.

Discussion

The discussion cannot be a repetition of the same concepts. The authors should start with the interpretation of the results, as the neglicence of employees, compare them with data obtained by other researchers, make hypotheses to explain the observations, and show how their results could be useful for other in similar or different situations. To do this, the authors need a knowledge of the literature, but at the moment there is absolutely no reference throughout the entire discussion.

Conclusions

These are too long and of limited interest.

Conclusions should be two or three phrases highlighting possible limitations, major findings and their importance.

Minor considerations

Lines 17-18: maybe “12.1 times higher” is better?

Lines 75-78: this phrase is not completely clear

Lines 117-119: this phrase is not completely clear

Lines 123-125: the observation reported here is of paramount importance. It should be commented thoroughly in the Discussion

Line 172: “working inside or outside”. Why are people working outside important for an in-hospital epidemic?

Line 199: what is the difference between confirmed and non-confirmed cases? Why there are non-confirmed cases? Have these been counted together with the confirmed cases?

Line 207: “P<0.001 was considered to indicate statistically significance”. Why  such a low P value and not P<0.05 as usual? And why in table 2 the authors used a P value <0.05? Moreover it is probably better to omit “statistically”

Lines 227 and lines 233-234: The authors say that confirmed cases were more frequent in males than females, but this could perhaps be due to higher number of male patients. The authors should give separately the percentage of positive  cases in men and women. Moreover a simple statistic  should be run to test significance; maybe a simple Chi square test or Fisher’s exact test is enough.

Lines 224-229: It is not clear if the majority of cases was aged 60 or more  in all the three Hospitals or only in two of them.

Lines 252 and 256: I cannot find Table 5. Maybe the authors refer to Table 2?

Lines 254 and 256: Omit “which was significant”: the P value is self-explanatory.

Table  2: In the caption there is a “**” symbol that I cannot find in the table

Dear Editor,

the manuscript needs major changes. Both introduction and discussion need to be rewrittwen. More references are needed.

Best regards

Round 2

Reviewer 1 Report

The revised manuscript is now acceptable.

Author Response

Thank you for your acceptable.
We'll do our best to finish.

Reviewer 2 Report

The manuscript still needs massive improvement.

The manner of writing is generally unprofessional. 

Author Response

Dear. Reviewer.

Overall English translation and edits to the summary and content.

Thank you.

Reviewer 3 Report

Introduction

Though new references have been added, the background of the study is still not enough outlined. What happened in other hospitals and in other countries? Have other types of epidemics occurred in previous years in the same or differnt hospitals? etc...

Moreover it seems to me that the introduction is quite too long.

Materials and methods

The information given in this section shuld be exposed in a shorter, more concentrated manner

Discussion

Though this has been changed it is still unsatisfactory. It cannot be accepted with not even a single reference: results should be discussed also in the light of the findings of other researchers.

Lines 128-129

I do not understand why privacy matters only for "unconfirmed patients".

The authors also say, in their previous answer to my comments that they " do not have information on the total number of unconfirmed cases and the gender of their members". But my question is different: How many inpatients  were males and how many were females and in each gender group how many were positive cases? To say that positive cases were more frequent in males we need this information

Lines 259-262

We cannot exclude that the six additional cases could have been infected before having a contact with the two index cases. This potential bias has to be discussed in the light of what is known in literature on the incubation period of the disease.

The manuscript needs phrasing and style editing by a native English speaker.
